# Good Practice Data Linkage (GPD): A Translation of the German Version [note 1]

**DOI:** 10.3390/ijerph17217852

**Published:** 2020-10-27

**Authors:** Stefanie March, Silke Andrich, Johannes Drepper, Dirk Horenkamp-Sonntag, Andrea Icks, Peter Ihle, Joachim Kieschke, Bianca Kollhorst, Birga Maier, Ingo Meyer, Gabriele Müller, Christoph Ohlmeier, Dirk Peschke, Adrian Richter, Marie-Luise Rosenbusch, Nadine Scholten, Mandy Schulz, Christoph Stallmann, Enno Swart, Stefanie Wobbe-Ribinski, Antke Wolter, Jan Zeidler, Falk Hoffmann

**Affiliations:** 1Institute for Social Medicine and Health Systems Research (ISMHSR), Medical Faculty, Otto von Guericke University Magdeburg, 39120 Magdeburg, Germany; stefanie.march@h2.de (S.M.); christoph.stallmann@med.ovgu.de (C.S.); Enno.Swart@med.ovgu.de (E.S.); 2Department of Social Work, Health and Media, Magdeburg-Stendal University of Applied Sciences, 39114 Magdeburg, Germany; 3Institute for Health Services Research and Health Economics, Centre for Health and Society, Faculty of Medicine, Heinrich-Heine-University Düsseldorf, 40225 Dusseldorf, Germany; Silke.Andrich@ddz.de (S.A.); andrea.icks@uni-duesseldorf.de (A.I.); 4Institute for Health Services Research and Health Economics, German Diabetes Center, Leibniz Center for Diabetes Research at the Heinrich-Heine-University Düsseldorf, 40225 Dusseldorf, Germany; 5TMF—Technology, Methods, and Infrastructure for Networked Medical Research, 10117 Berlin, Germany; johannes.drepper@tmf-ev.de; 6Techniker Krankenkasse, Healthcare Management, 22305 Hamburg, Germany; Dirk.Horenkamp-Sonntag@wineg.de; 7PMV Research Group, University of Cologne, 50931 Cologne, Germany; peter.ihle@uk-koeln.de (P.I.); ingo.meyer@uk-koeln.de (I.M.); 8Epidemiological Cancer Registry of Lower Saxony, Register Center, 26121 Oldenburg, Germany; kieschke@offis-care.de; 9Leibniz Institute for Prevention Research and Epidemiology—BIPS Department Biometry and Data Management, 28359 Bremen, Germany; kollhorst@leibniz-bips.de; 10Berlin-Brandenburg Myocardial Infarction Registry e. V., 10317 Berlin, Germany; birga.maier@herzinfarktregister.de; 11Center for Evidence-Based Healthcare (ZEGV), University Hospital and Faculty of Medicine Carl Gustav Carus, Technical University of Dresden, 01307 Dresden, Germany; Gabriele.Mueller@uniklinikum-dresden.de; 12IGES Institut GmbH, 10117 Berlin, Germany; christophohlmeier@gmail.com; 13Institute for Public Health and Nursing Research (IPP), University of Bremen, 28359 Bremen, Germany; dpeschke@uni-bremen.de; 14Department of Applied Health Sciences, University of Health Bochum, 44801 Bochum, Germany; 15Institute for Community Medicine, Department SHIP-KEF, Greifswald University Medical Center, 17475 Greifswald, Germany; adrian.richter@uni-greifswald.de; 16Central Research Institute for Ambulatory Healthcare in Germany (Zi), Department of Data Science and Healthcare Analyses, 10587 Berlin, Germany; MRosenbusch@zi.de (M.-L.R.); MaSchulz@zi.de (M.S.); 17Institute of Medical Sociology, Health Services Research and Rehabilitation Science (IMVR), Faculty of Human Sciences and Faculty of Medicine, University of Cologne, 50933 Cologne, Germany; nadine.scholten@uk-koeln.de; 18DAK Gesundheit, Health Services Research and Innovation, 20097 Hamburg, Germany; stefanie.wobbe-ribinski@dak.de (S.W.-R.); antke.wolter@dak.de (A.W.); 19Center for Health Economics Research Hanover (CHERH), Leibniz University Hanover, 30159 Hanover, Germany; jz@cherh.de; 20Faculty of Medicine and Health Sciences, Department of Healthcare Research, Carl von Ossietzky University Oldenburg, 26129 Oldenburg, Germany

**Keywords:** record linkage, guidelines, standard, personal data, health services research, epidemiology

## Abstract

The data linkage of different data sources for research purposes is being increasingly used in recent years. However, generally accepted methodological guidance is missing. The aim of this article is to provide methodological guidelines and recommendations for research projects that have been consented to across different German research societies. Another aim is to endow readers with a checklist for the critical appraisal of research proposals and articles. This Good Practice Data Linkage (GPD) was already published in German in 2019, but the aspects mentioned can easily be transferred to an international context, especially for other European Union (EU) member states. Therefore, it is now also published in English. Since 2016, an expert panel of members of different German scientific societies have worked together and developed seven guidelines with a total of 27 practical recommendations. These recommendations include (1) the research objectives, research questions, data sources, and resources; (2) the data infrastructure and data flow; (3) data protection; (4) ethics; (5) the key variables and linkage methods; (6) data validation/quality assurance; and (7) the long-term use of data for questions still to be determined. The authors provide a rationale for each recommendation. Future revisions will include new developments in science and updates of data privacy regulations.


**For the Working Group “Survey and Utilization of Secondary Data” (AGENS) of the German Society for Social Medicine and Prevention (DGSMP) and the German Society for Epidemiology (DGEpi), and for the Working Group on the ”Validation and Linkage of Secondary Data” of the German Network for Health Services Research (DNVF), as well as for the Working Group on Data Protection and the Working Group on IT Infrastructure and Quality Management of the TMF—Technology, Methods, and Infrastructure for Networked Medical Research e. V.**


## 1. Introduction

Existing guidelines for good scientific practice aim to achieve and maintain standards for conducting and appraising research. In doing so, they set different priorities, e.g., GUidance for Information about Linking Data sets (GUILD) [1], REporting of studies Conducted using Observational Routinely-collected health Data Statement (RECORD) [2], and Strengthening the Reporting of Observational Studies in Epidemiology Statement (STROBE) [3,4]. Experts from the above-mentioned working groups developed Good Practice Data Linkage (GPD) in Germany, which can be understood as a contribution to and further development of existing activities in the field of data linkage. GPD focuses on the administrative, logistical, and legal aspects of data use in addition to the primarily scientific planning, data processing, and reporting. This document was published in German in 2019 [5]. Although some aspects of GPD are specific to the situation in Germany, others can easily be transferred to an international context. As part of the European Union (EU), the General Data Protection Regulation (GDPR) [6] applies not only for Germany but also for the other EU member states, which builds one important legal framework for data linkage. Therefore, the authors would welcome an international exchange. In order to make these guidelines and national best practice examples available for a broader, international community, the authors decided to translate GPD into English and to publish the document as open access. Some elements specific to Germany have thus been excluded in this international version of GPD.

### 1.1. Objectives and Target Group of GPD

GPD establishes a standard for conducting research projects within the fields of health and social research and that seek to employ the data linkage of personal data according to scientific principles. Two primary objectives are pursued. First, it represents a manual that comprises guidelines for scientists who plan and carry out research projects with data linkage. While the guidelines and recommendations listed in the following will not be fully implemented in all research projects, this methodological standard requires critical reflection on the planned approach and a rationale for use in cases that deviate from the recommendations of this best practice. Second, GPD also serves as a checklist for the below-mentioned target groups in the evaluation of planned projects and in the appraisal of publications on such projects. In view of the increasing number of projects with data linkage and the associated methodological challenges, these institutions are thus provided with an instrument for ensuring sound feedback is provided to research groups. GPD aims to contribute to an improvement in the quality of research projects.

GPD complements the German guidelines and recommendations of the Good Practice in Secondary Data Analysis (GPS) (an older version of the GPS is available in English (https://www.dgepi.de/assets/Leitlinien-und-Empfehlungen/Practice-in-Secondary-Data-Analysis.pdf; accessed on 7 September 2020) of the DGSMP and the DGEpi [7] as well as the guidelines and recommendations for ensuring Good Epidemiological Practice (GEP) of the DGEpi; the German Society for Medical Informatics, Biometry and Epidemiology (GMDS); the DGSMP; the DNVF; and the TMF [8]. GPD is aimed especially at the following target groups:Data owners;Scientists;The reviewers of scientific projects and publications;Supervisory authorities, ethics commissions, and data protection officers.

Data linkage is often understood as merely the linking of various data sources. There are several synonyms such as record linkage, data matching, or entity resolution. However, we decided to use the broad term data linkage because some of the synonyms are also often used for specific parts of the process. We see data linkage as a comprehensive process involving the planning of a research project, the assessment, the actual merging of different data sources, and making the data available to third parties, for which the data have to be processed (e.g., deletion and anonymization)—in brief, all of the steps in the process that have to be considered in the application of data linkage. The guidelines provided here reflect all these steps. Merging identical data sources as part of a follow-up or data aggregation are not focused on. 

Therefore, the term “data linkage” is understood by the authors as a superordinate term for the overall process described above. In comparison, “record linkage” is understood as part of the overall process, i.e., as a toolbox for merging records, and thus describes the technical aspect of data linkage.

The data sources that are to be linked may be related to natural persons (hereinafter referred to as personal data) and therefore make it possible for those persons to be directly or indirectly identified. In addition, legal entities related to institutions (for instance, hospitals) may also be the subject of data linkage. 

### 1.2. Areas of Application

GPD exclusively discusses the linking of personal data from different data sources, irrespective of their origin or combination, i.e., all linkage possibilities for collections of primary and administrative data (in German, also referred to as secondary data or routine data) [9] containing personal data as defined above. The linkage of personal data with aggregated research data (aggregate data), such as classification systems, regional data, etc., is not considered in the context of GPD. Moreover, GPD deals with all of the process steps only insofar as they concern the topic of data linkage, and only provides recommendations for this purpose (General recommendations that go beyond the topic of data linkage (e.g., for general compliance with data protection regulations) are not provided. Reference is here made to the corresponding legal requirements or the GPS [7].). The primary focus of GPD is on the healthcare system with its specific (data) structures. Nevertheless, GPD can also be applied to other areas.

## 2. Methodology

In 2016, a “data linkage” project group met for the first time, initially consisting of eleven experts from the DGSMP and DGEpi working group AGENS and the DNVF working group on the Validation and Linkage of Secondary Data. As part of an inventory of German research projects with data linkage and their procedures, the Status Quo Data Linkage paper was published at the beginning of 2018 [10]. In May 2018, the kick-off meeting for the creation of a GPD took place in Hanover, at which the project group was expanded to 23 members.

In terms of content, GPD is based on the “Status Quo Data Linkage” publication (The “Status Quo Data Linkage” publication [10] contains, among other things, practical examples of German studies, concrete procedures and types of data linkage, a list of various software tools, specific information on quality assurance, and a checklist of the most important questions about data linkage.) [10] and represents its further refinement. Figure 1 visualizes the overall process of data linkage in the form of a flowchart and serves as a structured, process-oriented table of contents of GPD. Seven guidelines were defined, containing specific recommendations and references to further literature. In this way, the structure of GPD is in accordance with the other German “good practices” [7,8]. Specifically, GPD includes the following guidelines:The research objectives, research questions, data sources, and resources;The data infrastructure and data flow;Data protection;Ethics;The key variables and linkage methods;Data validation/quality assurance;The long-term use of data for questions still to be determined.

Good Practice Data Linkage is checked by the authors in cooperation with the above-mentioned working groups and other bodies in the course of regular scientific exchanges to ensure that it is up to date. Revision is carried out where necessary. 

### 2.1. Guideline 1: Research Objectives, Research Questions, Data Sources, and Resources

When formulating research questions and elaborating the research design, data sources that may be suitable should be identified by considering their potential and limitations, and the knowledge gain that is expected by means of data linkage should be described.

The research design must be oriented towards the research objectives or research questions formulated a priori. Thus, for each specific research question, an assessment of the planned collection of primary data and secondary data for linkage is necessary, and the potential and limitations of the individual data sources should be contrasted separately. In a summary assessment of all potentially suitable data sources, an appraisal of the expected additional knowledge to be gained through the linkage of two or more data sources should then be presented, weighing up the time and effort associated with the research project.

#### 2.1.1. Recommendation 1.1: Research Objectives, Questions, and, Where Applicable, Hypotheses Must Be Formulated as Precisely as Possible in Order to Create a Detailed Profile of the Requirements for the Data to Be Used

The research objectives and questions for research projects in which data linkage is foreseen must be precisely formulated since they provide the reference point for determining the specific requirements of different data sources. For this reason, it should first be shown whether there are only research objectives (e.g., in the German National Cohort (GNC) [11,12,13]) or if there are also specific research questions or hypotheses with deductive approaches involved (see GPS [7] and GEP [8]).

For research objectives and questions to provide a sufficient basis for the selection of suitable data sources or for the collection of new data, these must be small-scale and be able to be precisely operationalized. Only then will it be possible to specify the requirements for the data sources to be used with regard to the information they contain and the required quality of such information (see Recommendations 1.2 and 1.5). These are to be recorded in a data requirement profile (“What should the data be able to do?”), which forms the basis of decision-making for the selection, possible collection, and type of merging of the data to be used. A requirement profile is therefore a prerequisite for selecting suitable data sources in a targeted manner or, if necessary, designing new collections. This also indicates whether data linkage is necessary at all (see Recommendation 1.5). An entity relationship diagram needs to be created to formulate the research question and define the specific form of the data linkage.

#### 2.1.2. Recommendation 1.2: The Availability of the Information Needed to Pursue the Research Objectives and Answer the Research Questions Must Be Verified as Early as Possible. In Addition, the Research Design, Relevant Observation Periods, and Study Population Have to Be Specified

An important step in the planning, in addition to the specific content of the research objectives and research questions, concerns the selection of the data to be examined in order to answer the question. This involves, in particular, checking the availability of the required information and weighing up the possible need for the collection of primary data that may still be missing. When processing personal data, checking to see if this is in fact necessary is something that is required by law. The principle of data minimization must be complied with in this regard (Article 5 (1) (c) of the GDPR [6]). Accordingly, it must be assessed what information and which data sources are absolutely necessary in order to answer the research question in a meaningful way while adhering to the data minimization requirement under the law (see Recommendation 1.1). Particular attention is paid to the presence and quality of key variables in the various data sources, as they are a prerequisite for data linkage (see Guideline 5). Another important step concerns the definition and designation of the research design (e.g., cross-sectional, case-control, cohort, or validation study), the relevant observation periods, and the study population to be examined (on the basis of inclusion and exclusion criteria) (see GPS [7] and GEP [8]).

#### 2.1.3. Recommendation 1.3: From a Data Provenance Perspective, Data Owners Must Disclose and Communicate Developments in the Data Sources. In Particular, Historical/Systematic Changes to the Data Sources, Such as the Introduction of New Coding Systems or Key Variables, Must Be Communicated

Data sources are often subject to temporal changes and may thus involve systematic changes [15]; a new coding system may have been introduced, for example. Such systematic changes should be assessed, communicated, and documented for data linkage [16]. Ideally, this will also be documented in the entity relationship diagram.

#### 2.1.4. Recommendation 1.4: The Data Sources to Potentially Be Linked Must Be Described in Terms of Their Origin, Their Original Intended Use, Their Data Owner, and Their Advantages and Disadvantages

The data should be described in terms of their origin, their original intended use, their data owner, and their advantages and disadvantages. In principle, one can distinguish between two types of data for data linkage:Primary data, that are prepared and analyzed in the context of their original intended purpose, e.g., survey data.Secondary data, that are used beyond their original, primary intended use. These include, for example, a large number of data from social insurance institutions (e.g., statutory health and pension insurance), as well as other health care data (e.g., from medical office software or hospital information systems) or data from (clinical) research projects that are subsequently used to answer further questions. A more detailed description and examples of different types of data linkage can be found in Jacobs et al. [17], March et al. [10], and Swart et al. [18].

Additional data sources can also be used for data linkage. Details concerning these and other data sources in Germany can be found, among other places, in Antoni et al. [19], Antoni and Seth [20], Czaplicki and Korbmacher [21], Kajüter et al. [22], Korbmacher and Czaplicki [23], Maier et al. [24], March et al. [25], March [26], Ohlmeier et al. [27,28], Ohmann et al. [29], Swart et al. [30], and Stallmann et al. [31].

The description of the data sources or the detailed discussion of, among other things, the process of the creation of the data sources to be linked is considered necessary, as this can, for example, have an impact on the validity of the data contained and thus on the linkage variables potentially taken into account. Knowledge about the origin of the data should therefore be incorporated into the decision on the linkage procedure (e.g., fault-tolerant, yes/no; see also Guideline 5, Recommendation 5.3).

When assessing the individual data sources and their contents, related quality criteria such as completeness and an absence of errors [32] should be taken into account [33]. From an epidemiological point of view, quality criteria such as objectivity, reliability, and validity should also be considered. Depending on the structure of the data [9], the selection of supplementary quality criteria such as the context of the recording, topicality, personal and time references, and the possibilities of internal validation can serve to clarify the expected knowledge gain from an individual data linkage (see Recommendation 1.1) [33,34].

#### 2.1.5. Recommendation 1.5: The Use of Data Linkage Must Be Justified by the Question

Carrying out data linkage is justified if the question (see Recommendation 1.1) cannot be adequately answered without data linkage. If no further knowledge gain is to be expected through the use of data linkage, then this is to be refrained from for reasons of data minimization (see Guideline 4). Data linkage is useful if, for example, a data source alone does not contain all the required information, or if the information needs to be validated by another data source. For example, the limitations of a data source can be compensated for in this way by adding another data source.

If the data linkage is carried out with informed consent, attention should be paid to possible distortions due to non-response, as the risk of selection bias increases with each additional data source to be linked on account of the possibility that only a certain selection of study participants would agree to all of the mergers. Since this increasingly compromises the representativeness of the study, it is important to carefully weigh up the advantages and disadvantages of the data linkage. Restrictions may also occur if not all of the notified data owners provide data despite the consent of the participants (see March et al. for an example [35]).

#### 2.1.6. Recommendation 1.6: Due to the Complexity of a Research Project with Data Linkage, Sufficient Time, Financial, and Human Resources Must Already Be Provided for When Planning and Elaborating the Design

The planning of a research project with data linkage is complex. Sufficient time, financial, and human resources must be provided to define not only content-theoretical (pre-)considerations but also the general-practical requirements for conducting a research project. This includes the earliest possible involvement of data owners whose data are to be used for linkage. Any problems regarding data infrastructure and data flow (see Guideline 2) or data protection (see Guideline 3) can often be clarified or at least reduced beforehand. It is also necessary to check with the data owners’ data protection officers (institutions not part of the research process) how the data protection requirements can be met. It may also be necessary to schedule an ethics vote (see GPS [7] and GEP [8]) and make it available to the data owners.

When planning, it is necessary to take into account whether the research project with data linkage is to be conducted with or without the consent of the participants or whether a (supplementary) request for informed consent is possibly considered (see Guideline 3, Recommendation 3.2). 

The technical prerequisites for data management/data linkage should be estimated in the study planning and then compared with the existing conditions and resources. The timely purchase and commissioning of hardware and/or software that may be required can make a contribution towards the study running smoothly.

The early clarification of all the organizational, technical, and legal aspects helps to make a final decision for, or also possibly against, data linkage if the conditions necessary for data linkage cannot be met or the expected gain in knowledge through data linkage does not justify the effort.

Further examples can be found in the Status Quo Data Linkage publication [10], in Stallmann et al. [31] (GNC [36]), in March et al. [25,26] (lidA study), and in Swart et al. [37] (AGil study).

### 2.2. Guideline 2: Data Infrastructure and Data Flow

Data transfer and processing in a research project with data linkage take place on the basis of technical infrastructure, which has to fulfill certain conditions regarding its composition and the configuration of the individual components. This concerns the institutions involved in the data processing, data transfer, and data management.

For the institutions involved in data processing, their specific roles within the research project and the relationships between them must be defined, especially with regard to:The data to be exchanged and further information (e.g., data dictionary);Necessary mutual obligations where applicable (e.g., timely data deletion);The technical procedures used.

With regard to data transfer, projects with data linkage mean that transmission paths with more than one data owner and, if applicable, additional entities (a trust center, additional linkage quality assurance service, etc.; see Guideline 6) must be planned and described. When storing/managing data, the use of linkage software must be taken into consideration, as well as the potentially increased protection requirements of the newly created linked data, which may also be reflected in the data protection measures (e.g., in an appropriate security infrastructure).

In formal terms, the description of the infrastructure as well as the specifications and processes needed to meet the prerequisites referred to can be presented as part of a data flow concept. All of the elements are listed in a flowchart and described in greater detail in this regard.

A description of the data linkage should be given at the level of the variables, ideally starting with data dictionaries [38]. Entity relationship diagrams, for example, which are used to document database models, are suitable for the schematic representation of the data linkage [39]. In addition, the schematic representation enables the comparison of key variables (see Guideline 5) as well as the representation of variable properties (e.g., the data type).

#### 2.2.1. Recommendation 2.1: The Data Flow and Responsibilities Must Be Clearly Defined

With regard to the data processing units, it is initially advisable to separate the institutions involved in the research project from the roles to be performed in the project (e.g., data-gathering vs. data-processing vs. data-evaluating bodies). In this way, for example, an institution A can take care of the collection/provision of personally identifying data and the pseudonymization of said data (e.g., acting as a trust center), while institution B performs the linkage of the different data sources. The resulting separate views can help to achieve conceptual clarity and reveal any gaps that still exist in the process.

The following roles are typically found in projects with data linkage:Data-collecting agencies responsible for data acquisition;Various data owners (depending on the type of data supplied);A trust center, which pseudonymizes personal data, in particular;Bodies that anonymize personal data;Bodies that perform the data linkage themselves;Bodies that conduct linkage quality control;Bodies that perform the data evaluation.

Accordingly, data access has to be tied to these roles. Depending on the research project, the roles can be distributed among one or more institutions.

Data flow concepts from projects of varying complexity can be found, for example, in Hassenpflug and Liebs [40], Jacobs et al. [17], Pommerening et al. [41], and Swart et al. [37].

#### 2.2.2. Recommendation 2.2: The General Technical and Organizational Requirements for Data Transfer (See Guideline 6.1 of the GPS) Must Be Observed, and the Special Features of Projects with Data Linkage Have to Be Taken into Account 

It has to be clarified in fundamental terms whether the data used are transmitted from place to place as data records or whether at least some data are available only by way of data access (e.g., via a Virtual Private Network (VPN) or Remote Desktop Protocol (RDP)). Projects with data linkage are often complex projects in which special attention has to be devoted to the protection of data (see Guideline 3). For example, it must be ensured by means of suitable encryption and pseudonymization procedures that ensure impermissible data access (such as the combination of directly identifying data and medical data) cannot occur at any time. This can be achieved, for example, by multiple pseudonymization, by the separate encryption of key variables and medical data, or by the use of encrypted identifiers (e.g., Bloom filters; see Guideline 5).

#### 2.2.3. Recommendation 2.3: Software That Is Suitable for the Selected Record Linkage Method Must Be Used

Direct record linkage using existing, unique key variables can usually take place in the data processing software itself. For more complex procedures, March et al. [10] provide a detailed overview of currently available software for both exact and fault-tolerant linkage.

#### 2.2.4. Recommendation 2.4: A Suitable Process Must Be Defined for the Deletion of Data and for Contradiction Management 

On the one hand, the handling of the complete data set at the end of the research project has to be regulated. Depending on the planning, this can mean either the deletion or anonymization of the data set. This applies to the complete data set of all the participating institutions and roles in the flow of data, even in those institutions that conduct encryption or pseudonymization. On the other hand, special features of the IT infrastructure used must be taken into account. The latter are relevant, for example, where data also have to be deleted from data backups either directly or by cyclic overwriting.

For contradiction management, this means providing clearly defined processes for the deletion of individual data records or for the deletion of certain characteristics from individual data records (see also Recommendation 3.3).

Further information on sample studies can be found in the Status Quo Data Linkage publication [10] as well as in Jacobs et al. [17] and Swart et al. [37], among others.

### 2.3. Guideline 3: Data Protection

As explained in detail in Guidelines 1 and 2, the mere use of primary or secondary data sources may require special attention to be paid to data protection stipulations. When linking such data sources, a further increase in protection requirements can be expected. To this end, all relevant persons must be involved in the planning at an early stage. In addition to the actual data owners, these also include the internal or external data protection officers for the bodies involved and, where applicable, the competent supervisory authorities [10,25,26,41,42,43]. Depending on the data source, there are different specifications or applications to be made for use of the data [44].

#### 2.3.1. Recommendation 3.1: Data Protection Regulations Must Be Taken into Account during the Initial Planning Stage Right through to the Completion of the Project. The De-Anonymization/Re-Identification of Individual Persons through the Linkage Must Thereby Be Prevented

The more the data that are linked, the higher the risk that a natural person can be re-identified. In addition to data protection, the identification of legal persons can also affect other protection interests, e.g., competition-related information about doctors’ practices, hospitals, or health insurance companies.

It must be ensured that appropriate data protection measures are implemented at each point of transmission and processing. For this reason, procedures compliant with data protection provisions as well as the definition of responsibilities and competences for the processing, storage, and transport of data play a special role. Ideally, Standard Operating Procedures (SOPs) are formulated in relation to the following topics:Data protection and data security;Ethical and legal regulations concerning data access and use, where applicable;Structure and maintenance of the database(s);Data transfer and data deletion.

#### 2.3.2. Recommendation 3.2: It Must Be Checked Whether a Declaration of Consent Is Necessary

Research projects that use personal data usually require a declaration of consent, so-called informed consent. In the informed consent, the study participants must, after receiving adequate information, explicitly agree to the data linkage that has been planned and described [10,42].

It should also be noted that the use of information such as the pension insurance number or the life-long individual health insurance number usually requires the consent of the person concerned [31]. In some countries, there may be exceptions to the need for informed consent (see [10]), for example, when contacting study participants is impossible or unreasonable.

In addition, the informed consent must state under what conditions or when the data will be deleted. It must be ensured that the consent can be revoked at any time, which must generally lead to the complete deletion of all data concerning the revoking party that have not yet been anonymized.

These provisions regarding the scientific use of personal data and the linking thereof apply equally to research projects involving direct contact with the study participants. Even in these cases, consent must usually be obtained to link different personal data records.

In addition, the recommendations listed in 4.1 should be taken into consideration.

#### 2.3.3. Recommendation 3.3: A Data Protection Concept Must Be Developed

In addition, a separate data protection concept must be developed for each project. This concept sets out in writing the data flows and tasks, as well as the duties and responsibilities of all parties involved in the project [10]. The following detailed information must be included:A description of the project (background, objective, database, and methodology);Responsibilities (which public and non-public bodies are involved);The identification of the persons who have access to the data (trust center and researchers);The names of the persons involved, their data used, and/or the data categories (in particular, the key variables);The legal basis;Data-related processes, the resulting risks or protection requirements, and confidentiality;Organizational and technical measures or procedures;Time limits/deadlines;A concrete procedure for data deletion, including clarification of when data can/shall no longer be deleted, e.g., due to the anonymity of the data (see also Recommendation 2.4);Cancellation management: defining a procedure for deleting individual data records when requested to do so by a participant (see also Recommendation 2.4).

GPS [7], GEP [8], and the German Research Foundation (DFG) [45] recommend keeping the data for 10 years after the completion of the study. Accordingly, data should only be deleted after this time has elapsed, if possible (the “deletion concept” can be based on DIN 66398 (https://www.datenschutzbeauftragter-info.de/din-norm-66398-die-entwicklung-eines-loeschkonzepts/; accessed on 7 September 2020), which lists the points to be dealt with). The suitable storage of the data for this period must be ensured and, if necessary, should also be covered by a contractual agreement [7].

The MOSAIC project of the Institute for Community Medicine at the University Medical Center of Greifswald provides a template for drawing up a data protection concept (https://www.toolpool-gesundheitsforschung.de/produkte/vorlage-datenschutzkonzept; accessed on 7 September 2020) [46]. As part of the data protection concept, the extensive processing of health data requires a data protection impact assessment in accordance with the GDPR [6]. Generic concept guidelines, also agreed with all relevant data protection authorities, can be found in Pommerening et al. [41]. 

#### 2.3.4. Recommendation 3.4: If Linkage in a Research Project Is Only Planned Retrospectively, a Careful Examination of the Data Protection Regulations That May Have to Be Adhered to Needs to Take Place

If it is only during the course of a research project that a decision is made to link additional data to the existing data, the data protection officers of the participating institutions should also be contacted. In addition, it must be checked whether a linkage is possible in compliance with the present consent, as a later linkage may not be covered or may even be excluded from the outset. This can also affect indirect data linkage at a later stage; see, for example, [44]. It may subsequently be necessary to ask participants (again) to consent to the linking of their data. In this case, it must also be clarified whether there is even consent to contact the participants again, for the purpose of obtaining their consent to the linkage.

### 2.4. Guideline 4: Ethics

In addition to data protection, the use of linked data sources usually also has an impact on the ethical evaluation of the research project and on any professional advice required by the ethics committee responsible or by a use and access committee involved on the basis of other regulations (see Guideline 7). These concern, in particular, the scientific value, the quality, and, where applicable, also the originality of the project if new insights are created by the merging of data that have so far only been analyzed separately or not at all. Under certain circumstances, the same arguments may also establish increased practical relevance. The combination of different data sources can thus have an impact on the benefit–harm potential of the project, whereby the potential harm caused by a possibly greater re-identification potential due to the merging of the data can conflict with the potential increase in benefit resulting from the new findings.

#### Recommendation 4.1: Possible Effects of the Data Linkage on the Benefit–Harm Potential of the Research Project Must Be Examined

When performing data linkage, the following aspects must be considered from an ethical perspective:The minimization of incorrect linkage and any resulting false outcomes;Minimizing the risk of the re-identification of natural persons (see Guideline 3).

With respect to the risk of re-identification, an ethical discussion should be held on the use of data-storage mechanisms that allow the merging of different data sources only at the time of analysis and only with the data that are imperative for the analysis (see Guideline 7). When drawing up information for participants and consent, care must be taken to describe each data source (data category) separately. The information for the participants must be designed in such a way that every single person is able to understand the nature and effects of the linkage.

### 2.5. Guideline 5: Key Variables and Linkage Methods

Key variables and the various procedures used to link data records play a central role in data linkage. The selection and use of key variables and linkage procedures are interrelated and must therefore be coordinated.

The term key variables refers to those variables that occur in all the data records to be linked and thus facilitate assignment. Linkage methods are technical procedures used to link the data sources via key variables. After linking the data records, it should be checked whether the key variables are still required or if they can be deleted in the new data record (data minimization principle).

#### 2.5.1. Recommendation 5.1: Before Defining and Using Key Variables, the Existing Framework Conditions Regarding Their Use for Linkage Must Be Clarified

The selection of a suitable linkage method, including the key variables to be used, depends on different framework conditions:The legal (data protection) requirements to be observed must be clarified. Of particular importance is the question as to whether the linkage ensues on the basis of informed consent, as this provides the framework for the available key variables. For the purposes of risk assessment, it should be examined whether linkage increases the risk of the re-identifiability of individuals (see Guideline 3).The type of data source influences the quality of the key variables. In contrast to in data that are collected retrospectively, in prospective data, it is possible, where applicable, to supplement (collect) variables that enable or simplify subsequent linkage.It has to be clarified at what points in time the record linkage is to take place: automatically at the time of data collection, at regular time intervals (e.g., per quarter), or after the final acquisition of all the data sources for the research question. Particularly in the case of long-term projects such as registries or the establishment of research databases, the timing of the data collection and data consolidation should be described. The variables to be collected, including the key variables, must also be defined.

#### 2.5.2. Recommendation 5.2: All Key Variables Must Be Precisely Defined and Checked with Regard to Their Completeness and Susceptibility to Errors

The quality and completeness of the data linkage are largely determined by the key variables available. Special attention should therefore be paid to their selection and description.

Automatically recorded variables that can be used as keys (e.g., specific insurance numbers) are to be given priority (if possible). The use and comparison of the check digits reduces linkage errors that can arise due to key variables being collected or transmitted incorrectly.It must be checked to what extent directly identifying identifiers (e.g., names or insurance numbers) are used as key variables in plain text form or are to be masked by suitable procedures (pseudonymization, a hash function, or a Bloom filter). It should be taken into account to what extent the selected masking method can be used for each data source. Depending on the existing framework conditions and data protection requirements, this masking can be carried out in the same way by any data owner whose data are to be linked (possibly using a pseudonymization service such as Mainzelliste [10,47]) or with the involvement of a trust center/data trustee [41]. Further information on this topic can be found in the Status Quo Data Linkage publication [10].Appropriate procedures should be used to minimize false negative (synonym errors) or false positive classifications (homonym errors). In this way, different spellings of the key variables can be harmonized (e.g., phonetic coding methods, substrings, and Bloom filters [48,49,50,51]) and assignment errors reduced by incorporating further features. If key variables can change over time (e.g., married names), appropriate precautions must be taken for further assignment (e.g., a translation table for translating old to new IDs or the inclusion of the maiden name—see also Recommendation 6.2).

#### 2.5.3. Recommendation 5.3: A Suitable Technical Procedure Must Be Chosen for the Data Linkage

Several steps are required to conduct the technical process of linkage, e.g., data cleaning, the selection of candidate pairs, the comparison of all candidate pairs, and the review of matches. A detailed explanation of these steps is given in Vatsalan et al. [52]. Within the step “comparison of candidate pairs”, different methodological approaches can be applied. 

A distinction is made between direct and indirect linkage and between probabilistic and deterministic linkage. In addition, there are exact and fault-tolerant methods, which are, in turn, subdivided into rule-based and distance-based fault-tolerant methods. Furthermore, blocking methods [53], in which only data sets with the same manifestations of specific features are compared, can also play an important role because they are able to improve the performance of the linkage process; however, they can also adversely affect the quality of the linkage.All processes have advantages and disadvantages that must be considered before they are used. They can be linked together under certain conditions. It is advisable to involve IT officers and experts in the choice and implementation of the linkage process at an early stage.If the records to be linked contain plain-text identifiers or pseudonymized plain-text identifiers as key variables, direct linkage methods can be used. However, the use of pseudonymized identifiers is only possible if the data records to be linked contain key variables that have been pseudonymized according to the same procedure.If the result of the direct linkage is not satisfactory, e.g., because data records could not be linked due to incorrect key variables, then an additional indirect procedure can be used (see Guideline 6). Deterministic record linkage with indirect identifiers is considered an indirect procedure for data linkage. If direct patient identifiers such as names are not available, data may be linked through a combination of indirect identifiers, i.e., age, sex, and point in time [54].

If the data records to be linked do not contain any key variables, it can then be checked to what extent the data records contain identical variables that can be used as key variables and thus enable indirect linkage.

Further information and examples of studies can be found in the Status Quo Data Linkage publication [10].

### 2.6. Guideline 6: Data Validation/Quality Assurance

Aspects to ensure data quality must be considered as part of the planning and preparation of the data linkage process. The need for such activities is not limited to a separate step in the process for data linkage but, rather, applies starting from the quality assurance of the individual data records, to the actual record linkage process, and right through to the verification and quality assurance of the linked data set, i.e., for all of the data linkage steps to be taken.

In addition to the technical requirements, the planning should include sufficient human resources for preparing the data and also enabling the verification of the success of the data linkage. After the data linkage has been carried out, for example, a comparison with the transferred data may be necessary [55] (see also Recommendation 6.3) or it may be necessary to code the clear text data of clinical findings in a standardized way prior to an evaluation.

#### 2.6.1. Recommendation 6.1: A Description of the Quality of the Key Variables Must Be Included in the Project Report

The initial quality of the key variables is a crucial factor for the success of the linkage. The validity of the key variables can differ significantly in different data sources, e.g., manually recorded personal information from handwritten death certificates vs. electronically transmitted mortality data from residents’ registration offices. In addition, it may be necessary to include data fields in the plausibility checks that are not part of the key variables (e.g., age, sex, weight, or height) if they are used to detect incorrect assignments (for an overview, see [56]). If possible, a validation study should be conducted to assess the quality of the key variables. Finally, the data linkage procedure and the result of the data linkage should be explicitly described and evaluated in the research project report.

#### 2.6.2. Recommendation 6.2: It Must Be Examined Whether an Iterative Procedure Leads to Better Linkage Quality

The proportion and type of incorrect record linkage results (synonym or homonym errors) depend, among other things, on the overlap of the same persons in the records to be linked, the probability of randomly matching key variables, and the likelihood of differing information for the same person (except for errors such as a change of name after marriage, change of residence after moving house, etc.). Estimates should be made in relation to these parameters. As this information is often not yet known when planning a study, it may not be possible to finalize the details about the data linkage at that time. The iterative approach as well as the proposed procedure and the maximum tolerable rate of mismatches should nevertheless be specified beforehand, e.g., in the study protocol.

#### 2.6.3. Recommendation 6.3: In the Context of Quality Assurance, It Must Be Possible for the Body Conducting the Evaluation or the Body Performing the Data Linkage to Make Inquiries with the Data Owner. Implausibilities Must Be Clarified with the Data Owner to Avoid Inconsistent Data or a False Interpretation of the Data

As a rule, a direct and immediate inspection of the data owners’ unencrypted data cannot be carried out for reasons of data protection, especially in the case of social data. For this reason, arrangements should be made in the contractual provisions between the body conducting the evaluation and the body performing the data linkage regarding the form in which the data owner can provide assistance in clarifying queries concerning the original data. In the case of ambiguity, for example, the body conducting the evaluation or performing the data linkage should ask the data owner the appropriate (detailed) questions concerning the data set, who can then carry out a check within the (non-pseudonymized) original data. Depending on the result of the check, further validation steps can be planned, or, if necessary, a modified new data set may be made available. Such results and appropriate adaptation measures could, for example, also be the result of a pre-planned validation study to determine the synonym and homonym error rate.

#### 2.6.4. Recommendation 6.4: After Each Data Linkage, the Number of Merged and Non-Mergeable Records Must Be Checked on the Basis of the Source Files

For this purpose, an estimate is to be made in advance as to how often successful links would have to occur for the individual files. In addition, the observed frequency distributions should be checked for plausibility after the linkage has been completed. It should be checked whether the non-linked records display a specific structure that is either responsible for the failed link (e.g., source A is up to date whereas the record is not yet available in source B) or that could cause systematic selection bias in the linked data set. For this reason, the essential variables in the linked and underlying non-linked data sets should be examined to verify structural differences.

#### 2.6.5. Recommendation 6.5: After Each Data Linkage, a Comparison Must Be Made between the Transferred and the Merged Data

Merging data sources from different operating systems or software applications can result in undetected errors or distortions, or even a loss of data. For example, (i) special characters can be misrepresented by an incorrect coding, e.g., the storing and reading of character encodings can be an issue; (ii) there may be data types/formats that are either non-existent or misinterpreted in the target system; or (iii) due to limited field lengths, truncations of the character length may be made in variables. A comparison between the transferred and aggregated data can be made on a random basis to check for such distortions or losses. Insofar as systematic changes have been made to a data source over time, this data reconciliation should take these different episodes into account [55].

#### 2.6.6. Recommendation 6.6: The Actual Error Rate Must Be Measured and Included in the Result Report. If the Linkage Is Repeated Several Times, the Error Rate Must Be Continuously Checked and Compared with Previous Results

If a linkage is scheduled several times, e.g., where there are new data every year, the error rate can also change over time. On the one hand, this can be due to dynamic data, such as data from a cancer registry, which are constantly updated. On the other hand, it can provide indications of possible implausibilities in the data or indicate an error that has not yet occurred. This quality assurance process must therefore be carried out again and documented for each linkage (see Recommendation 6.1).

#### 2.6.7. Recommendation 6.7: After Each Data Linkage, the Description of the Properties of the Resulting Research Data Set Must Be Made with Reference to the Original Data

The data sources to be linked might not match the number of observations (e.g., subjects, study participants, and insured persons). Successful data linkage thus implicitly represents a selection process, which can lead to distorted results [57]. Each form of observation selection should be described, and an estimation should be made of the associated changes in the properties as compared to the original data.

### 2.7. Guideline 7: Long-Term Use of Data for Questions Still to Be Determined

Data linkage is a complex process and generates complex bodies of data with a wealth of information. These often display potential for analysis that goes beyond the evaluation goals of research projects that run for a limited period of time. In addition, some research projects have the aim, right from the outset, of building up a research database with extensive data sets that are available long term. In this respect, further data use by authorized third parties may be considered under a priori-determined conditions if this is feasible in accordance with the data protection regulations.

#### 2.7.1. Recommendation 7.1: If the Data Owner Intends to Make Further Use of the Merged Data beyond the Primary Issue, or Should This Possibility Exist in Principle, the Appropriate Regulations Must Then Be Taken into Account as Early as at the Design Stage of a Research Project

With the linkage of different data sources, the resulting data pool usually has considerably more research potential than simply enabling an answer to the primary question. For this reason, considerations already need to be made in the conception phase about whether and in what form the data will be used. If, as a result of these considerations, it is determined that it should be possible to use the data later in a more open manner, this must be mentioned explicitly in the ethics application as well as in the data protection concept and in the contractual arrangements with the data owners, and it must also be covered by the contract concluded with the data owners. In the case of consent-based data linkage, the informed consent must contain a flexibility clause that legitimizes the use of the data for possible further questions and explains the conditions under which further use is planned.

#### 2.7.2. Recommendation 7.2: If the Merged Data Are to Be Made Accessible for Scientific Use by Third Parties within the Framework of a Research Database, This Use Must Be Regulated by a Standardized Access Procedure

Numerous research projects in epidemiology and social science aimed at answering research questions that are not restricted in terms of content and time are designed from the outset to establish a research database geared to this purpose. These databases are intended to facilitate the scientific use of the collected data by those not directly involved in the research projects.

This planned use of research data obtained in a time-consuming and laborious manner requires a standardized and transparent application and approval procedure, e.g., in the context of a so-called use and access arrangement, with a Use and Access Committee being responsible for the decision on an application for use [41,58]. It should be determined from the outset under what conditions data usage is possible and which body examines these conditions and grants or denies appropriate clearance and release.

Examples of use and access arrangements of this kind in Germany are provided by population-based research projects such as the nationwide GNC or regional surveys and cohort studies (e.g., Study of Health in Pomerania (SHIP study)) (current versions of the use and access arrangements can be found on the project websites: https://nako.de/allgemeines/der-verein-nako-e-v/rechtliche-grundlagen/; http://www2.medizin.uni-greifswald.de/cm/fv/ship/datennutzung/; accessed on 7 September 2020) as well as the medical informatics initiative of the Federal Ministry of Education and Research (BMBF) (https://www.medizininformatik-initiative.de/sites/default/files/inline-files/MII_03_Use_and_Access_Policy_Key_Issues_Paper_1-0.pdf; accessed on 7 September 2020).

The research projects also make it possible to offer ready-made data bodies from the already merged primary and secondary data in the form of scientific use files. In order to safeguard data protection and also ethical as well as professional and technical interests [41,58], a standardized and transparent procedure for access to the data for third-party researchers should also be established here. Even with anonymized data, it should be borne in mind that these may retain linkage potential, meaning that indirect linkage then remains possible.

This is of particular importance if the data to be disclosed are going to be linked to other data bodies, thus increasing the risk of the re-identification of study participants through the greater depth of information. Risk variables that potentially support such indirect linkage with other available data sets as key variables should therefore be deleted wherever possible or their content sufficiently simplified.

## Figures and Tables

**Figure 1 ijerph-17-07852-f001:**
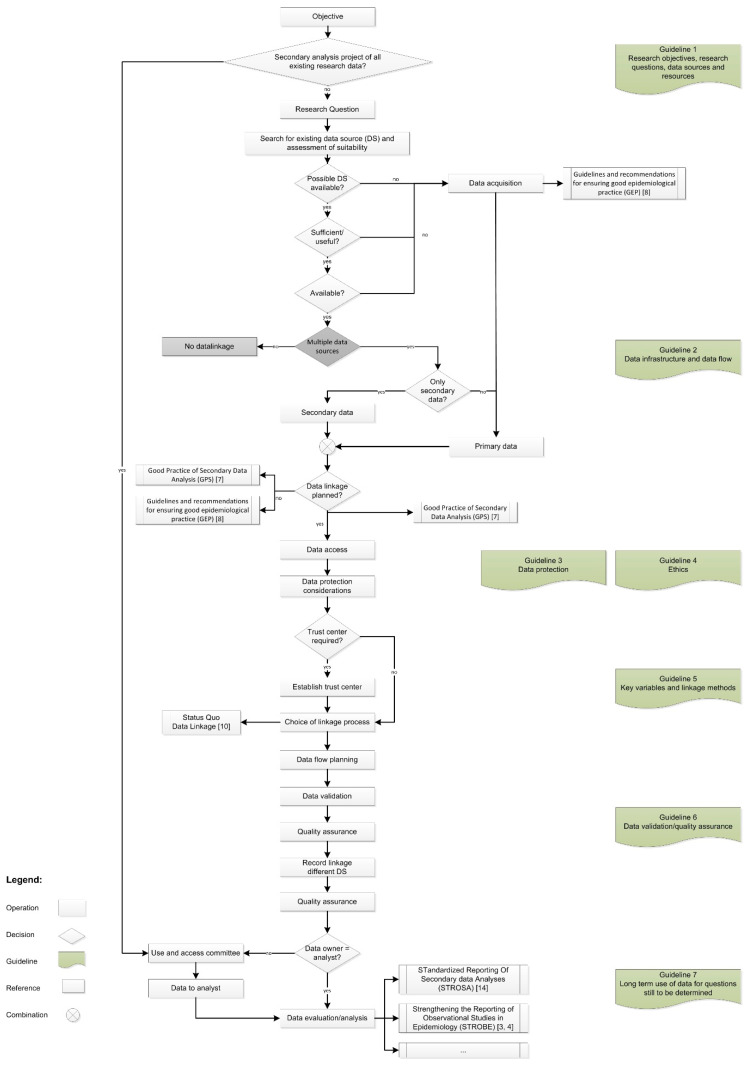
Overall process of a research project with data linkage. Strengthening the Reporting of Observational Studies in Epidemiology (STROBE) [3,4]; Good Practice in Secondary Data Analysis (GPS) [7]; Guidelines and recommendations for ensuring good epidemiological practice (GEP) [8]; Status Quo Data Linkage [10]; STandardized Reporting Of Secondary data Analysis (STROSA) [14].

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
