# Peer review of "Good Practice Data Linkage (GPD): A Translation of the German Versionâ€"

_ijerph, 2020, doi:10.3390/ijerph17217852_

Round 1
Reviewer 1 Report
Many thanks for the opportunity to review this manuscript. I understand that this is a translation of an already published paper, and this manuscript is aimed at an international audience.
I note below areas that I feel should be addressed to make this accessible to an international audience as well as some comments on the general readibility. This manuscript is very long and can definitely be reduced in length whilst still retaining the important content.
- The glossary could go into an appendix or at least reduced to include terms specific to this paper that the authors feel need defining. E.g. Aggregate data, pseud/anonymisation, check digit, consent, data dictionaries, DPIA, Key variable, personal data, primary/secondary data, selection bias - These are all used in papers without a definition usually so not sure what it adds here to define them as the definitions are not new. Data owners = data providers and/or data controllers as defined by GDPR? (if we're making this internationally accessible) Is Legal person - data controller under GDPR?
- Figure 1 "further use of data planned outside the research project" - yes - doesn't make sense to me - perhaps re-word if it's specfically asking if this is a secondary analysis project and not required to design a study
- Figure 1 - Data protection considerations should be at the start (privacy by design) and not after data access (or is data access in the wrong place?)
- International context but only German examples of e.g. Data sources. - is it for those accessing data in Germany only?
- What is partial informed consent?
- line 258 - Data owners' data protection officers - often it is the data access team of the data provider
- line 260 - Ethics vote - don't reference but elaborate here (or provide as a definition)
-
line 388 - Concept - not sure this the right term, a plan? Protocol?
-
What about the Ethics of onward data sharing? ensuring it is within reasonable expectations?
- No mention of public / participant involvement which is of particular importance when considering ethics.
- line 499 - Procedure for linkage (isn't this record linkage and not data linkage?) - not always possible as it's up to data providers quite often in the UK this needs to be recognised.
-
There is cross over with GUILD (https://academic.oup.com/jpubhealth/article/40/1/191/3091693) and RECORD statement (https://www.record-statement.org/) - these need to be read and referenced appropriately and explained where they diverge if addressing international audience.
Minor comments / typos to be addressed:
- Footnotes should be incorporated into the text or as references
- "On the one hand" could be re-worded to be more formal (line 76 & 81)
- Committee (typo) in figure 1
- Data Transmission = data transfer?
- GDPR and GDPR terms applicable to Europe but not beyond...
- References are sometimes in square brackets but other times superscript
Author Response
Dear editor,
please find below our detailed response to the comments of the reviewer. In the article itself, all changes are highlighted in yellow (as already references were modified, we did not use the track change mode to make the document clearer).
One general aspect: The article was professionally translated into English as well as further revised by a professional lector and native speaker (see also the acknowledgement).
With kind regards,
Stefanie March & Falk Hoffmann (for the authors)
________________________________________________________________
Reviewer 1:
1) This manuscript is very long and can definitely be reduced in length whilst still retaining the important content.
The text was reduced in length, especially the glossary was deleted as suggested (see our response to comment 2).
2) The glossary could go into an appendix or at least reduced to include terms specific to this paper that the authors feel need defining. E.g. Aggregate data, pseud/anonymisation, check digit, consent, data dictionaries, DPIA, Key variable, personal data, primary/secondary data, selection bias - These are all used in papers without a definition usually so not sure what it adds here to define them as the definitions are not new. Data owners = data providers and/or data controllers as defined by GDPR? (if we're making this internationally accessible) Is Legal person - data controller under GDPR?
We thank the reviewer for this suggestion and decided to delete the glossary from the manuscript (as the other reviewers also commented on this).
3) Figure 1 "further use of data planned outside the research project" - yes - doesn't make sense to me - perhaps re-word if it's specfically asking if this is a secondary analysis project and not required to design a study
Good point, figure 1 was modified accordingly.
4) Figure 1 - Data protection considerations should be at the start (privacy by design) and not after data access (or is data access in the wrong place?)
We have discussed the right position of data protection in figure 1 when creating this figure as well as now. As there are only “theoretical” questions at the beginning of the flow chart, we decided to insert the aspect of data protection exactly on this position. We do not think that a possible project should start with thinking about data protection before research questions, infrastructures and issues on which data will be linked are clearly elaborated.
5) International context but only German examples of e.g. Data sources. - is it for those accessing data in Germany only?
We modified the introduction to make this point clearer. In order to make GPD available for a broader, international community, we decided to translate the GPD into English and to publish the document as open access. We would like to go on an international exchange, especially in Europe because of common rules like the GDPR. Therefore, we focused on national best practice examples in which we are involved, because of the national experience of the author team.
6) What is partial informed consent?
We deleted the word “partial”.
7) line 258 - Data owners' data protection officers - often it is the data access team of the data provider
In Germany, the data owners, e.g. hospitals, statutory health insurance funds, are not regularly part of the research process. That´s why these institutions have their own “data owners' data protection officers”. The paragraph was, therefore, modified.
8) line 260 - Ethics vote - don't reference but elaborate here (or provide as a definition)
We decided that general recommendations like ethics vote that go beyond the topic of data linkage are not further discussed, because it is part of the research process in general.
9) line 388 - Concept - not sure this the right term, a plan? Protocol?
The article was professionally translated into English and further revised by another professional lector for medical literature (who was also involved in the Good Epidemiological Practice). Therefore, we continue to use the word “concept” as we also believe that terms like “plan” or “protocol” might lead to confusion.
10) What about the Ethics of onward data sharing? ensuring it is within reasonable expectations?
It is not common in Germany due to rules of data protection or contracts with data owners.
11) No mention of public / participant involvement which is of particular importance when considering ethics.
We totally agree, these aspects are very important in research. However, the GPD focused on technical aspects of data linkage and legal requirements that have to be fulfilled. That´s why only the mentioned working groups and other bodies in the course of regular scientific exchanges participated. As public / participant involvement is part of the research process in general and specific for data linkage, it is not further discussed.
12) line 499 - Procedure for linkage (isn't this record linkage and not data linkage?) - not always possible as it's up to data providers quite often in the UK this needs to be recognised.
We changed the introduction of recommendation 5.3 to make it more comprehensive and clearer.
13) There is cross over with GUILD (https://academic.oup.com/jpubhealth/article/40/1/191/3091693) and RECORD statement (https://www.record-statement.org/) - these need to be read and referenced appropriately and explained where they diverge if addressing international audience.
We added the references in the introduction. We decided only to reference these papers, because of the large variety of guidelines. Furthermore, we added an explanation on the aim and focus of our work to clarify potential differences.
Minor comments / typos to be addressed:
14) Footnotes should be incorporated into the text or as references
As the reviewer mentioned, the manuscript is quite long and we decided to put some additional information in footnotes in order not to enlarge the text.
15) "On the one hand" could be re-worded to be more formal (line 76 & 81)
We re-worded “on the one hand” as suggested.
16) Committee (typo) in figure 1
The spelling was corrected in figure 1.
17) Data Transmission = data transfer?
The word “data transmission” was replaced by “data transfer”.
18) GDPR and GDPR terms applicable to Europe but not beyond...
We decided to focused on Europe, because of common rules like the GDPR. We now make this focus more clear in the introduction. Nonetheless, several aspects might also be applicable to other countries.
19) References are sometimes in square brackets but other times superscript
We checked the manuscript again: All references are in square brackets, but footnotes, which also include references, are in superscript numbers.
Reviewer 2 Report
In scientific literature and practice, the concept of data linkage has several synonyms: record linkage, data matching, entity resolution, etc. The manuscript would serve a more wide audience if the terminology was spelled out in the beginning. The authors do mention on page 6, row 104, that “record linkage” is only one part of the overall process and “data linkage” is a superordinate term. Anyway, giving a plain definition of the concept of data linkage would be most welcome. On page 6, rows 97-102, the concept is described, but in a circular manner: “data linkage is a comprehensive process involving … all the steps that have to be considered in the application of data linkage”. Instead, definitions are given for example in Wikipedia https://en.wikipedia.org/wiki/Record_linkage , by medical research institutes https://www.menzies.utas.edu.au/research/research-centres/data-linkage-unit/what-is-data-linkage or by Eurostat https://ec.europa.eu/eurostat/cros/system/files/s-dwh-m_4.2_methodology_data_linkage_v2.pdf I also wonder if “data aggregation” is also a synonym for data linkage? https://en.wikipedia.org/wiki/Data_aggregation Please clarify.
Page 2, row 46: “Therefore, it was now also published in English” – if this manuscript is now the English version then the sentence would better read as “Therefore, it was now also published in English”.
In the Glossary in Table 1, pages 3-6, several entries in fact do not give a definition of the term in question but merely describe some aspects of the term. Please find coherent and self-contained definitions.
Page 3, Table 1, last entry: Please either give a definition for consent / informed consent or remove this entry.
Page 4, first entry “Data dictionaries”: This definition slightly differs from a well-known one https://en.wikipedia.org/wiki/Data_dictionary Instead, the definition given here, “In a data dictionary, all variables of a data record should be listed and their properties described. These properties include the name, possibly a label, the data type (e.g. numerical, string, date, date-time stamp) and an explanation of the contents of the variables [10]” sounds similar to “metadata”. Please explain the relation between this definition and the definition of metadata, and/or give a definition for “metadata” as well.
Page 4, second entry “Data minimization”: Please give a definition. It is in general not the same as data economy.
Page 4, fifth entry “Data provenance”: What is data-aiding? Another comment: The description provided in this entry is not in fact a definition of the term “data provenance” which by e.g. Wikipedia is “records of the inputs, entities, systems, and processes that influence data of interest, providing a historical record of the data and its origins” https://en.wikipedia.org/wiki/Data_lineage and other definitions are given in https://link.springer.com/referenceworkentry/10.1007%2F978-0-387-39940-9_1305
Page 4, seventh entry “Entity relationship diagrams”: This description is almost impossible for me to understand. Please provide a definition of the concept.
Page 5, second entry “Legal persons”: is this the same as a legal entity? Please clarify.
Page 6, Table 1, last entry: please write this description in the form of a definition.
The English language of the manuscript is often complicated and sometimes the sentences are long, making the text hard to understand at first. I wonder if the authors could ask a native English speaker to check the language. Examples: page 7, rows 147-149; page 10, row 195; page 11, rows 275-277; page 12, rows 285-287; page 16, rows 473-475; page 19, rows 626-628;
Page 9, Figure 1: Please make the font size bigger in this figure, or re-draw it to better fit into one page.
Page 16, row 503: While probabilistic linkage is a well known concept, it seems to be out of the scope of this manuscript, as far as I understood – but I might be mistaken. My hunch is that many aspects described in the earlier chapters only apply to deterministic linkage. I wonder if the authors could explain probabilistic linkage in more detail, at least stating which recommendations apply to probabilistic linkage and which do not. Alternatively the whole concept of probabilistic linkage could be omitted from this manuscript.
Page 17, row 517: Please explain in more detail what you mean by indirect procedures, or please indicate which aspects of Guideline 6 this refers to.
Re-use of the data in other contexts is only briefly mentioned. This is understandable in light of privacy constraints, as the main field of application is health records. However, the concept of FAIR data (findable, accessible, interoperable and reusable) https://en.wikipedia.org/wiki/FAIR_data https://www.force11.org/group/fairgroup/fairprinciples https://www.openaire.eu/how-to-make-your-data-fair is gaining more and more popularity in the scientific community, and also among research funding organizations and publishers. It would be good to at least briefly touch upon this concept in the manuscript.
Author Response
Reviewer 2:
20) In scientific literature and practice, the concept of data linkage has several synonyms: record linkage, data matching, entity resolution, etc. The manuscript would serve a more wide audience if the terminology was spelled out in the beginning. The authors do mention on page 6, row 104, that “record linkage” is only one part of the overall process and “data linkage” is a superordinate term. Anyway, giving a plain definition of the concept of data linkage would be most welcome. On page 6, rows 97-102, the concept is described, but in a circular manner: “data linkage is a comprehensive process involving … all the steps that have to be considered in the application of data linkage”. Instead, definitions are given for example in Wikipedia https://en.wikipedia.org/wiki/Record_linkage , by medical research institutes https://www.menzies.utas.edu.au/research/research-centres/data-linkage-unit/what-is-data-linkage or by Eurostat https://ec.europa.eu/eurostat/cros/system/files/s-dwh-m_4.2_methodology_data_linkage_v2.pdf
The reviewer is right, there are several synonyms in the literature. We modified the paragraph accordingly and integrated the synonyms.
25) I also wonder if “data aggregation” is also a synonym for data linkage? https://en.wikipedia.org/wiki/Data_aggregation Please clarify.
We clarified the paragraph in “Merging identical data sources as part of a follow-up or data aggregation are not focused on.”
26) Page 2, row 46: “Therefore, it was now also published in English” – if this manuscript is now the English version then the sentence would better read as “Therefore, it was now also published in English”.
Unfortunately, the reviewer stated the same sentence twice. Therefore, we are unclear what should be changed.
27) In the Glossary in Table 1, pages 3-6, several entries in fact do not give a definition of the term in question but merely describe some aspects of the term. Please find coherent and self-contained definitions.
As also reviewer 1 criticized on this point, we decided to delete the glossary from the manuscript. As the following comments also apply to definitions in table 1, we reference to this comment in our answers to comments 28 to 34. All of them refer to the definitions given.
28) Page 3, Table 1, last entry: Please either give a definition for consent / informed consent or remove this entry.
See comment 27, the glossary was deleted.
29) Page 4, first entry “Data dictionaries”: This definition slightly differs from a well-known one https://en.wikipedia.org/wiki/Data_dictionary Instead, the definition given here, “In a data dictionary, all variables of a data record should be listed and their properties described. These properties include the name, possibly a label, the data type (e.g. numerical, string, date, date-time stamp) and an explanation of the contents of the variables [10]” sounds similar to “metadata”. Please explain the relation between this definition and the definition of metadata, and/or give a definition for “metadata” as well.
See comment 27, the glossary was deleted.
30) Page 4, second entry “Data minimization”: Please give a definition. It is in general not the same as data economy.
See comment 27, the glossary was deleted.
31) Page 4, fifth entry “Data provenance”: What is data-aiding? Another comment: The description provided in this entry is not in fact a definition of the term “data provenance” which by e.g. Wikipedia is “records of the inputs, entities, systems, and processes that influence data of interest, providing a historical record of the data and its origins” https://en.wikipedia.org/wiki/Data_lineage and other definitions are given in https://link.springer.com/referenceworkentry/10.1007%2F978-0-387-39940-9_1305
See comment 27, the glossary was deleted.
32) Page 4, seventh entry “Entity relationship diagrams”: This description is almost impossible for me to understand. Please provide a definition of the concept.
See comment 27, the glossary was deleted.
33) Page 5, second entry “Legal persons”: is this the same as a legal entity? Please clarify.
See comment 27, the glossary was deleted.
34) Page 6, Table 1, last entry: please write this description in the form of a definition.
See comment 27, the glossary was deleted.
35) The English language of the manuscript is often complicated and sometimes the sentences are long, making the text hard to understand at first. I wonder if the authors could ask a native English speaker to check the language. Examples: page 7, rows 147-149; page 10, row 195; page 11, rows 275-277; page 12, rows 285-287; page 16, rows 473-475; page 19, rows 626-628;
The article was professionally translated into English by one person as well as revised by another professional lector and native speaker (see also the acknowledgement). As is was translated and checked twice by different native speakers, we already did our very best concerning the English language.
36) Page 9, Figure 1: Please make the font size bigger in this figure, or re-draw it to better fit into one page.
We now tried our best to change the figure, but we have to use the template of the journal. However, the article will only be available open access. After publication there will also be a html-version of this paper and the figure will be available online, which makes it easier to enlarge respective parts.
37) Page 16, row 503: While probabilistic linkage is a well known concept, it seems to be out of the scope of this manuscript, as far as I understood – but I might be mistaken. My hunch is that many aspects described in the earlier chapters only apply to deterministic linkage. I wonder if the authors could explain probabilistic linkage in more detail, at least stating which recommendations apply to probabilistic linkage and which do not. Alternatively the whole concept of probabilistic linkage could be omitted from this manuscript.
We redraft the introduction of recommendation 5.3 to make it more comprehensive and clearer. Due to lots of main technical points of fault-tolerant methods, there is limited space to discuss this topic. Moreover, the recommendation to use one technique or another depends on the available data sources. Therefore, additional information of data sources is given.
38) Page 17, row 517: Please explain in more detail what you mean by indirect procedures, or please indicate which aspects of Guideline 6 this refers to.
We modified the paragraph and added to clarify “Deterministic record linkage with indirect identifiers is considered an indirect procedure for data linkage. If direct patient identifiers like names are not available, data may be linked through a combination of indirect identifiers, i.e. age, sex and point in time.”
39) Re-use of the data in other contexts is only briefly mentioned. This is understandable in light of privacy constraints, as the main field of application is health records. However, the concept of FAIR data (findable, accessible, interoperable and reusable) https://en.wikipedia.org/wiki/FAIR_data https://www.force11.org/group/fairgroup/fairprinciples https://www.openaire.eu/how-to-make-your-data-fair is gaining more and more popularity in the scientific community, and also among research funding organizations and publishers. It would be good to at least briefly touch upon this concept in the manuscript.
Data privacy considerations are a major concern. This makes it difficult to re-use the data in another context, as for example the FAIR data (findable, accessible, interoperable and reusable) concept suggests. Nevertheless, the aspect of FAIR data should be considered for future data linkage, if possible or imaginable, taking into consideration the special data privacy rules for health data. However, this discussion is beyond the scope of our manuscript and, therefore, we decided not to include the concept of FAIR data.
Reviewer 3 Report
Dear Authors,
Thank you for this effort to translate the guidelines which describe in great detail efforts which constitute good practice for data linkage.
If possible, it would be valuable to describe and illustrate examples common pitfalls or grey areas where the guidelines may help to remedy. I understand that this may not then 100% reflect the German guidelines.
I would also suggest a list of knowledge gaps which are not covered by the current version of guidelines which then can be addressed in future updates of these guidelines.
Author Response
Reviewer 3:
40) If possible, it would be valuable to describe and illustrate examples common pitfalls or grey areas where the guidelines may help to remedy. I understand that this may not then 100% reflect the German guidelines.
In the before mentioned “status quo”, which was a previous version on the long road to a good practice, we included a large bulk of examples and pitfalls because we agree that these are very helpful. However, this was not the scope of the GPD, which includes guidelines and recommendations. In order to keep the text as short as possible (reviewer 1 and 2 also criticized that the text is too long), we decided not to include further examples here.
41) I would also suggest a list of knowledge gaps which are not covered by the current version of guidelines which then can be addressed in future updates of these guidelines.
Please see our response to comment 40. Because the text is already quite long, we also decided against a conclusion.
Round 2
Reviewer 1 Report
Many thanks for the response and justification for the suggested changes.